# Fluorescence-Based Protein Stability Monitoring—A Review

**DOI:** 10.3390/ijms25031764

**Published:** 2024-02-01

**Authors:** Negin Gooran, Kari Kopra

**Affiliations:** Department of Chemistry, University of Turku, Henrikinkatu 2, 20500 Turku, Finland; negin.gooran@utu.fi

**Keywords:** protein stability, fluorescence, intrinsic fluorescence, SYPRO Orange, thermal shift assay (TSA), isothermal chemical denaturation (ICD), Protein–Probe, urea, melting temperature (Tm), differential scanning fluorimetry (DSF), external dye, Gibbs free energy

## Abstract

Proteins are large biomolecules with a specific structure that is composed of one or more long amino acid chains. Correct protein structures are directly linked to their correct function, and many environmental factors can have either positive or negative effects on this structure. Thus, there is a clear need for methods enabling the study of proteins, their correct folding, and components affecting protein stability. There is a significant number of label-free methods to study protein stability. In this review, we provide a general overview of these methods, but the main focus is on fluorescence-based low-instrument and -expertise-demand techniques. Different aspects related to thermal shift assays (TSAs), also called differential scanning fluorimetry (DSF) or ThermoFluor, are introduced and compared to isothermal chemical denaturation (ICD). Finally, we discuss the challenges and comparative aspects related to these methods, as well as future opportunities and assay development directions.

## 1. Introduction

Proteins are involved in every aspect of physiology, and to correctly play their role, they need to be folded in a specific way. The structure and stability of these macromolecules are important factors to preserve their activity and function. Keeping the structure of a protein in a stable state is crucial in drug development, drug delivery, and fundamental research [1]. The structural stability of a protein is directly related to the environmental conditions [2] and buffer formulation, which vary from one protein to another. These factors relate to protein structure and amino acid sequence, defining how compact the protein is and what, for example, the isoelectric point (pI) is. These parameters thereafter determine the optimum storage conditions by defining the optimal pH, ions, and temperature. The structure of a protein under different conditions and with different additives should be studied to provide insight into its function and optimize the conditions. This is especially important in the case of biologics [3,4]. Protein engineering facilitates protein therapeutics, which can be categorized based on the type of the molecule [5]. The largest and fastest-growing subsection is antibody-based drugs. In this class of proteins, monoclonal antibodies (mAbs) are receiving attention from major pharmaceutical and biotechnology companies. Most of the mAb polypeptide chains can fold into a specific shape, which is critical for the functionality and stability of the protein. Keeping and delivering the mAbs in the correct shape is one of the major focuses of the pharmaceutical R&D section [6], and this relates not only to protein stability but also to its tendency to aggregate [7].

There are both label-free and label-based methods to study proteins, their structural stability, and interactions. Förster resonance energy transfer (FRET), and especially time-resolved FRET (TR-FRET)-based techniques utilizing two labeled macromolecules, is one of the most extensively used examples of label-based methods [8]. The disadvantages of these methods are high costs due to the labeling step needed for every target protein studied, and sometimes, the use of two labels might change the protein function, or the assay is difficult to construct [9,10,11]. Thus, label-free methods have gained greater interest, especially due to their flexibility and suitability for different protein targets without extensive optimizations. Label-free methods are generally easier to use and have greater accuracy than label-based techniques in the context of protein stability [12]. Thus, the most ideal method should be operated in a label-free manner, to detect minor changes in the stability of protein, and enable automation by measuring numerous samples synchronously [13]. However, there are numerous analytical parameters that regulate the success of a method such as the sensitivity, detection limit, high-throughput capability, and range of applicability [14]. For example, to be able to detect small changes in protein stability, the assay needs to have high sensitivity and a suitable resolution [15,16].

Even still, most protein stability assays are performed in a highly controlled manner using a single protein in a buffer, and different types of approaches are used. The cellular thermal shift assay (CETSA) is a practical example of using the principle of a label-free method in drug development in a cell environment [17]. In this method, the shift in the melting curve indicates the binding of the designed drug with the target protein [18]. On the other hand, methods creating high amounts of data, like mass spectrometry-based thermal proteome profiling (TPP), have gained interest, which gives insight at a proteome-wide level. TTP-type methods are like CESTA and can provide information regarding the state of the protein and interactions, which is used to recognize the off-target effects of the drugs [19,20,21]. Another practical example of CESTA is to determine the stability of the membrane protein without detergents. The other methods use detergents to make the membrane protein soluble, but the presence of the detergent affects the background [22,23].

In vitro label-free protein stability assays, such as differential scanning fluorimetry (DSF), have gained popularity, especially due to their simplicity and low instrumental demands [24,25]. DSF is based on adding an external probe fluorophore with a low fluorescence signal in a polar environment, such as in an aqueous solution, and monitoring fluorescence signal increases when this dye enters more nonpolar environments upon protein denaturation. DSF has many advantages over the gold-standard method for protein stability, differential scanning calorimetry (DSC) [26,27]. DSC can provide high-quality information regarding protein thermal stability and interactions between proteins and other molecules, but its throughput is limited. Similar to many other methods used for stability assessment, DSC is equipment-demanding, not having a similar applicability to various types of laboratories as DSF [28]. In this review, we briefly introduce some of the equipment-demanding techniques for protein stability monitoring, but the main focus is on non-equipment-limited methodologies with high-throughput (HT) capabilities. We look at the application of the methods, such as studying protein–ligand interaction (PLI) and formulation, basic principles, and the parameters, and discuss future trends, especially those related to DSF and isothermal chemical denaturation (ICD) techniques.

### 1.1. Usability of Protein Stability Assays

Analytical techniques like DSC have a low throughput, but in the case of DSF, the throughput is already increased. Low-throughput methods cannot fill the void of an assay for screening the immense number of compounds collected in pharmaceuticals [29], but their advantages are elsewhere. Thus, the selection of the correct method is of high importance. Research that grants multiple measurements of molecules simultaneously with the possibility of repetition has high throughput. HT methods are usually performed in 96- to 1536-well plates, and these methods have been extensively used for the last few decades [30,31]. HT techniques are well suited for biophysical characterization, buffer optimization, and studying PLIs and protein–protein interactions (PPIs). They are usually easy to perform and automate, with low demands on expertise [32]. One of the leading measurement methods in HT screening are fluorescence-based assays. They are favored due to primitiveness, their ability to adapt, high sensitivity, and low to zero damage to the target while analyzing it [29].

In the context of protein stability assays, HT methods can be categorized in two ways: (1) aiming to measure the protein properties under physiological conditions (e.g., 37 °C) and interpret the stability based on the measurements which are more favorable for studying human diseases at molecular levels, and (2) incomplete distressing and disturbing of the protein to measure the stability accurately [33]. There is research regarding developing an HT method independent of the target function and/or enzymatic activity to widen its range of applications, as the central point of failure of these approaches is the limited range of applications [34]. In the following part, we introduce some of the most important stability parameters and introduce the usability in the context of interaction monitoring utilizing protein stability assays. We also give a quick look at the methods with low throughput and high equipment demands, but those methods are more thoroughly introduced elsewhere [28,35,36,37].

### 1.2. Stability Parameters

There are several important parameters which can help scientists not only to interpret their results, but also to plan their research. Before the intended assay, one should have some prior knowledge about the studied protein or interaction. This information often relates to the protein environment, e.g., the buffer pH, structural features in secondary structures, aggregation properties, and so on [38,39]. By having information about the protein and given conditions, more reliable thermodynamic parameters can be obtained.

Among the thermodynamic parameters of protein unfolding, the Gibbs free energy of unfolding (ΔG) is a precise measure of protein stability. ΔG indicates the stability of the protein. A positive, large ΔG means that a protein is stable. The melting temperature (T_m_) is also used to show the stability of the protein [40,41]. At T_m_, the ΔG of the folded and unfolded states are equivalent, and the Tm defines the approximate temperature at which the protein is 50% folded. In thermal unfolding events, other parameters like T_on_, the beginning of the unfolding event, and T_turb_, the beginning of aggregation, can be given [42]. When ΔG is paired with T_m_, one can extract the enthalpy of unfolding (ΔH) and the entropy of unfolding (ΔS). These parameters are related to each other—(−RTlnK) = ΔG = ΔH − TΔS—where R is the gas constant, K is the equilibrium constant, and T is the absolute temperature [43]. All of these parameters are often used in the context of methods like DSC and DSF, and there are several papers in regard to thermodynamic properties and their relationship to each other, which is outside the scope of this review paper [44,45,46,47,48].

### 1.3. Interaction Monitoring

Protein stability assays are most often utilized for protein formulation and interaction monitoring. In particular, PLI assays are performed in all types of laboratories, from basic research in academia to screening in pharma. Among the factors that may influence protein stability are generic ingredients like buffers, salts, and detergents, whose interactions with the protein are not always specific for a single pocket in the protein structure. These factors are the main interest in formulation, as ligands that bind to the protein at a specific site have different types of effect. In addition to their effect as protein stabilizers, PLIs can be of high value for functional studies such as substrate specificity and for identifying allosteric effectors that would help in providing better protein annotations. Moreover, a treasured starting point in drug development is ligand identification [49]. Formulation and PLI assays also have different demands on the assay. Following the growth of the usage of proteins in the pharmaceutical industry and the importance of understanding their structure, the analysis of molecular interactions between ligands and their target molecules is currently in the spotlight, and the number of methods is also increasing [50]. Understanding the mechanisms of the interaction of proteins and ligands can speed up the process of drug discovery and development, and depending on the method used, the resolution of the gained information varies significantly [51,52]. X-ray crystallography, NMR, and cryo-electron microscopy are a few examples of the many experimental techniques that can be used to investigate PLIs in high resolution [53,54,55]. They can provide atomic-resolution or near-atomic-resolution structures of the unbound proteins and the protein–ligand complexes, which can be used to study the changes in structure and/or dynamics when binding happens, along with between relevant free and bound forms [56]. However, a high resolution is not always needed, as protein-level resolution might give the needed information [57].

The binding affinity may not be correctly determined without a complete thermodynamic profile and thermodynamic properties. This information cannot be achieved using only the structural and dynamic data, no matter how accurate and practical the computational methods are [58]. There is a need for a technique that can provide quantitative thermodynamic data that can be used to study the complex stability and elucidate the binding driving forces. DSC is one of the methods that can predict the stability of protein–ligand complexes via measuring the enthalpy and the heat capacity of thermal denaturation, but its usage in biochemistry laboratories is limited due to its expansive instruments that can only be used to denature proteins. Fluorescence spectroscopy-based techniques enable the reliable study of PLIs due to their high sensitivity and relative simplicity [59]. In this method, the fluorescence of the solution is monitored while the solution is heated, and when the protein chain begins to unfold, the hydrophobic core becomes exposed, leading to an increase in the signal until all protein molecules are fully denatured. Generally, the protein stability is increased by ligand binding, meaning that the T_m_ value increases [60]. Ligand binding often causes a relatively low shift in T_m_ (ΔT_m_), as in the case of the well-studied model protein, carbonic anhydrase [61]. As ligand binding-induced changes in T_m_ might be very small, methods must be able to reliably measure these changes. Depending on the method, a ΔT_m_ over 1 °C can be counted as significant, but not all interactions give this high of a change. However, especially some covalent binders and other high-affinity interactions might cause a ΔT_m_ of 10–30 °C or even higher, as in the case of the streptavidin–biotin pair [62,63]. These extreme stabilizers can cause their own problem, related to the high temperatures needed for denaturation. Other disadvantages, such as a relatively high protein consumption, which hides functional data in assays with high-affinity ligands, might cause interferences with some proteins and ligands with a limited usability at high concentrations, for example, because of their low solubility. Thus, new external dyes for protein unfolding are constantly developed [64,65,66].

Numerous protein targets and protein-based drugs have been identified in recent times, with an anticipation that approximately a dozen therapeutic proteins per year will receive regulatory approval in the upcoming decade [67]. For effective product development, the formulation scientist must have a good understanding of the mechanism of degradation of the macromolecule of interest and its potential impact on such areas as its biological activity, metabolic half-life, and immunogenicity [68]. Before evaluating safety, toxicity, absorption, distribution, metabolism, excretion (ADME), and pharmacology and conducting assessments of biological activity in animals, it is imperative to conduct protein formulation screening [67]. Fluorescence spectroscopy, particularly in combination with microscopy, is a powerful tool often used in a spectacular manner to study biological processes occurring in living cells [69]. In the field of therapeutic protein development, fluorescence spectroscopy is one of the most rapidly advancing areas [70]. One of the important steps in mAb formulation is the determination of the thermal unfolding of the antibody. This can readily be undertaken with the help of polymerase chain reaction (PCR) instruments. Instead of detecting the presence of amplified nucleotides, a fluorescence dye is used to detect binding associated with the appearance of solvent-exposed hydrophobic regions after unfolding [69].

### 1.4. Non-Luminescent Methods

Not all methods have equal suitability for all proteins and areas of interest. In addition, not all methods are available for researchers, and here, we give a short introduction to more equipment-demanding techniques for protein stability. The most used method to destabilize proteins of interest is heat. In these thermal stability methods, the sample is heated and the changes in experimental stability parameters are recorded. Thereafter, the properties and quantities of the native and denatured proteins, and potentially other stability parameters, can be determined using these records. The gold-standard technique in thermal stability analysis is DSC (Figure 1A) [71]. Other label-free methods that are often used are circular dichroism (CD) (Figure 1B) and nuclear magnetic resonance (NMR) spectroscopy. The common aspect of these three techniques, as well as multiple other applicable methods, like mass spectrometry (MS) and isothermal titration calorimetry, is the need for specialized instruments and scientists with expertise to perform these assays and interpret the results [72,73].

DSC is a dominant technique and can provide information on the unfolding process of macromolecules in terms of the structure, as well as changes in the protein environment and ligand binding (Figure 1A). This method is flexible and can be used for all soluble proteins, unlike DSF, in which unwanted dye binding to the native protein might affect measurements, but also for membrane proteins. DSC provides the excess heat capacity of macromolecules as a function of temperature; as a related technique, ITC measures the temperature change during ligand titration [74]. These two methods are the only ones that directly determine the ΔH. DSC enables the collection of valuable information, including (i) the absolute partial heat capacity of a molecule, (ii) the comprehensive thermodynamic parameters linked to a temperature-induced transition, and (iii) the partition function. Additionally, it facilitates the simultaneous determination of the population of intermediate states and their corresponding thermodynamic parameters. Generally, DSC cannot be used for small molecules, except when used as a protein ligand. Usual DSC measurements can be performed using low mg amounts of protein, and a typical scan takes 10 to 60 min, depending on the scan and equipment used [75,76].

CD is a well-established method to swiftly analyze the secondary structure of the protein (Figure 1B). It also provides insight for changes that happen during biological processes such as folding and interactions. This method needs a 0.05–0.5 mg/mL concentration of protein. CD falls under the category of absorption spectroscopy, as distinct structural elements show different absorption in the left- and right-circularly polarized light. Intrinsic CD counts as a label-free method since its probe is a part of the protein structure that can absorb light, called chromophores [77,78]. Examining the thermodynamic parameters linked to protein unfolding induced by heat, osmolytes, denaturants, or ligands through CD is achievable. CD finds diverse applications, including assessing the integrity of membrane proteins during extraction. Typically, CD is more time-consuming than DSC, taking a few hours to collect the data, but with the new and more advanced CD equipment, the assay can be performed in even less than an hour. CD can be used to give an estimate of the secondary structural composition of proteins, but it does not give similar residue-specific information to X-ray crystallography and NMR [79].

NMR is the second method besides X-ray diffraction in single crystals to determine protein structures at an atomic resolution. NMR provides data that are compatible with X-ray crystallography, thus helping to better understand the relation between structure and function. NMR spectroscopy is an important technique for studying time-dependent phenomena, including reaction kinetics and the intramolecular dynamics of macromolecules, along with weak protein–protein interaction, as it does not require crystallization [80,81]. To mention a few applications of NMR, we can point out the investigation of protein conformation changes, weak protein–protein interactions, and denaturation [81,82]. On top of these, NMR has the ability to quantify dynamics under equilibrium conditions without external perturbations, using many probes simultaneously and over large time intervals [83,84]. However, NMR studies are performed only in special situations needing high-resolution data, as the analysis takes a significant amount of material, time, and expertise [85].

## 2. Luminescence-Based Thermal Shift Assays (TSAs)

An ideal assay for universal use for varying sizes of laboratories is simple, cost-efficient, and does not need specialized instruments. Many of the fluorescence-based assays fill these criteria, and thus are often used when highly accurate and special information for protein stability is not needed. The thermal shift assay (TSA), also referred to as DSF or ThermoFluor^TM^, is a cost-effective, parallelizable, practical, and accessible biophysical technique (Figure 2). Thus, it is widely used as a method to track both the protein folding state and stability, as well as interactions [31]. DSF is a method that involves incubating naturally folded proteins with environment change-sensing fluorescent dye in a multiwell plate. As the temperature gradually increases, the fluorescence emission of the dye is monitored in real time. These recordings discriminate the properties and populations of native and denatured conformers. The introduction of fluorescence-based TSA has significantly enhanced the ease of identifying conditions that enhance protein stability, as most of the used external dyes are compatible with commonly available real-time PCR machines. In comparison to most of the other methods, DSF is fast (20–120 min) [86]. It also can be used for multiple things, e.g., to identify stabilizing conditions, additives, and small-molecule ligands for purified recombinant proteins along with cofactors. Data collected can be directly used to study protein properties, but also to indicate favorable crystallization or storage conditions or aid in assigning functionality. The TSA’s high-throughput nature allows for the quick discovery of protein-stabilizing solutions through the sparse matrix screening of various solution conditions, such as the buffer identity, solution pH, and ionic strength. Typically, DSF assays are performed in 20 µL volumes in 96- or 384-well plates, with 5–10 µM of protein in the final volume, and requires minimal specialized instrumentation [87]. Given these advantages, DSF is used widely in laboratories in academia, the government, and industry alike. To pinpoint a candidate that is capable of binding with the target protein while disregarding enzymatic activity, high-throughput DSF screens are employed. Additionally, these screens prove invaluable in constructing chemical libraries for drug development and formulating biotherapeutics [46]. The cause of many misfolding diseases, including cystic fibrosis, Gaucher’s disease, and Fabry’s disease, are genetically destabilized proteins [88].

The exploration of small-molecule correctors is possible via DSF. DSF has been applied to overcome the challenges of sample preparations Notably, DSF has been adapted to furnish data even in unpurified chemical reactions, exemplifying its utility in complex solutions [31]. Further enhancements to the methodology have resulted in the usage of dyes with superior spectral properties, analyzing the data with generic tools, and recommended protocols for preliminary screening [49]. Although DSF has a wide range of application in studying PLIs (Figure 2B), just a limited number of studies have successfully determined dissociation constants for the investigated PLIs [89,90]. These studies often involve detailed equations describing protein unfolding, necessitating the fitting of numerous parameters to sparse data or, in some cases, estimation. The importance of such techniques shows itself in the study of tightly binding compounds or proteins exhibiting unconventional transitions.

In this section, different types of intrinsic or external TSA dyes, the effect of used buffers for assay optimization, and equipment demands are explained. In addition, a brief summary of analyzing TSA data and a comparison of the results, accuracy, and sensitivity of TSA with other methods are given.

### 2.1. Fluorescent Dyes vs. Intrinsic Fluorescence

In all DSF-type assays, the first and most important step is to select a suitable readout method. The specialized fluorescent dyes often used in DSF are defined as compounds that both absorb light and emit strongly in the visible region (Figure 3) [91]. The fluorescent dyes suitable for DSF exhibit high fluorescence intensity when in a nonpolar environment, such as the hydrophobic sites found on unfolded proteins. In contrast, their fluorescence is quenched in an aqueous solution. The optical properties of the dyes may vary, especially in terms of the fluorescence quantum yield [92], which is affected by their binding to denatured proteins [93]. In some label-free methods, the more complex external probe is used to detect the structural integrity and interactions of the target protein. In addition, the intrinsic fluorescence of a studied protein might be applied [2].

Commercial dyes such as 4,4′-bis(phenylamino)-[1,1′-binaphthalene]-5,5′-disulfonic acid dipotassium salt (bis-ANS) and Nile Red (Figure 3B) have been used for decades, despite the fact that, in the presence of a folded protein, they might have a high signal background [31]. SYPRO Orange (Figure 3A) is the most favorable dye for DSF thanks to its high signal-to-background (S/B) ratio. Its spectral properties make it suitable for qPCR equipment, as its excitation is 488 nm and its emission range is 500–610 nm [31]. These wavelengths also are less prone to interferences, as most small molecules typically have absorption maxima at shorter wavelengths. In comparison, many compounds interfere with the spectral properties of another commonly used DSF dye, 1-anilino-8-naphthalene sulfonate (1,8-ANS), whose excitation maximum is at ∼350 nm (Figure 3C). Even 1,8-ANS has a number of advantages over some other probes, e.g., good solubility, a relatively small effect on protein structure and stability, convenient fluorescence properties, and suitable affinity for most proteins, excitation in the UV area is a major shortcoming ruling out most of the qPCR equipment [94]. Still, ANS together with SYPRO Orange are usually the first probes to be tested [94]. The GloMelt^TM^ kit, which includes GloMelt^TM^ dye, is commercially available for determining protein stability through TSAs, with an excitation of 468 nm and an emission of 507 nm. The advantage over SYPRO Orange is that GloMelt™ dye is slightly more sensitive, but especially that these assays can be performed at high detergent concentrations. Despite all the advantages of SYPRO Orange and other typical DSF dyes, not all proteins can be analyzed using these external probes [49]. As an illustration, SYPRO Orange, Nile Red, and 8-ANS all attach to hydrophobic regions within the protein. In contrast, Thioflavin T (ThT) and similar dyes preferentially bind to the “beta-pleated stack” and related motifs found in misfolded, refolded, or aggregated protein species formed during the unfolding process [95]. ThT and its modifications, used, for example, in the ProteoStat Kit, are rarely used to measure protein stability directly, but for aggregation monitoring (Figure 3D). The so-called “molecular rotor” microviscosity probe, e.g., 9-(dicyanovinyl)julolidine (DCVJ) and 9-(2-carboxy-2-cyanovinyl)julolidine (CCVJ), has also been applied in DSF monitoring (Figure 3E) [95]. Their extreme sensitivity, structural specificity, and versatility make extrinsic probes useful for high-throughput screening, especially related to formulation studies [95,96]. These dyes can tolerate detergents and surfactants even better than GloMelt™ dye, but due to the different mechanism of action, these dyes do not monitor T_m_ similar to dyes like SYPRO Orange, but rather protein aggregation properties (T_agg_) at higher temperatures [97]. A new class of external probes for aggregation and stability studies are the Protein–Probe family of peptide-targeted time-resolved luminescence (TRL) probes [64]. In these methods, one or two peptides are used to monitor proteins, and the advantage of these techniques is their extreme sensitivity at low nanomolar levels of protein. Three variations of this method have been published, all utilizing different detection principles and varying suitability [26,64]. Unfortunately, due to the TRL readout, none of these methods can be monitored using qPCR equipment.

External probes must always be added to the protein solution, and thus, there is always a possibility for interferences due to the used dye. Another source of fluorescence originates from the protein sample itself. Intrinsic protein fluorescence deriving from the naturally fluorescent amino acid tryptophan (excitation at ∼280 nm, emission at ∼350 nm) [98], and, to a lesser extent, from low-quantum-yield phenylalanine and from often-quenched tyrosine (excitation at ∼275 nm, emission at ∼304 nm) [99], can provide information on the conformational changes of proteins [100]. Tryptophan emission proves to be especially sensitive to the polarity of the local environment, making it suitable for reporting on the local-specific conformational changes as the protein unfolds [101]. The usual observation related to the unfolding is for the fluorescence emission maximum to undergo a red shift (toward a longer emission wavelength, from 330 nm to above 350 nm). This corresponds to the increased exposure of the tryptophan groups to the solvent in the unfolded state. Tyrosine fluorescence is generally only used for proteins that do not contain tryptophan, and it affects the assay sensitivity [102]. Thus, intrinsic fluorescence is most powerful for proteins with tryptophan by following protein unfolding using progressive redshifts on tryptophan fluorescence emission due to the changes in the indole ring microenvironment [103,104]. NanoDSF (nDSF) is a modern method that is also used for monitoring changes in intrinsic tryptophan fluorescence. Unfortunately, these assays are still out of reach for many scientists, as the specialized devices used for nDSF are still not universally found and their usage is quite expensive in comparison to conventional DSF [105].

Even though intrinsic fluorescence and external probes are the most widely used techniques due to their label-free nature, sometimes, target protein labeling might be a rational option. Schaeffer’s lab introduced an innovative method that utilized a green fluorescent protein (GFP) to quantitatively assess the stability of a target protein [31]. In these experiments, a GFP tag was fused to the targeted protein through a peptide linker, establishing a reporter system for tracking the unfolding and aggregation of the protein. The fluorescence signal emitted by the GFP was responsive to its immediate surroundings, allowing it to be employed for tracking the unfolding process of the linked protein. At around 75 °C, the GFP starts to lose fluorescence; hence, the stability of the GFP is higher than a lot of proteins [31]. The method further developed into a GFP thermal shift (GFP-TS) assay, which is a form of chromatography, excluding the size, as a streamlined adaptation of fluorescence detection. The hybrid approach integrates the adaptability of the sample that is inherent in the above-mentioned chromatography, with the high-throughput capacity of dye-based thermal shift assays. GFP-TS, in this context, proves effective for discerning specific ligand interactions within solute carrier transporter fusions and quantifying their affinities in crude detergent-solubilized membranes. Also, purified SLC transporter fusions can be investigated by GFP-TS in order to discover specific lipid–protein interactions. Similar to nDSF, spectral shift assays have also been used with labeled proteins [106]. By attaching a specific spectral shifting dye, sensing the microenvironment, to the studied protein, the assay affinity can be significantly increased to a sub-nanomolar level, similar to the Protein–Probe approach. The use of near-infrared dyes does not need to be site-specifically conjugated to the target, but only the labeling degree must be optimized. Target protein labeling is especially useful for those targets that are typically difficult to study, and in those cases, assay costs often do not limit the research.

As there are often multiple areas of interest, especially with mAb drugs, lately, some combinatory systems have been created. One of these systems is the Uncle protein stability screening platform [107]. Uncle boasts three detection methods—full-spectrum fluorescence, static light scattering (SLS), and dynamic light scattering (DLS)—to profile protein stability. With full-spectrum fluorescence detection (250–720 nm), protein intrinsic fluorescence and dyes like SYPRO Orange can be scoped to assess protein unfolding and denaturation parameters. Simultaneously, SLS tracks the formation of large and small aggregates, while DLS takes care of sizing and size distributions [108]. This type of combinatory system increases the speed of protein analysis, as multiple parameters can be obtained simultaneously. However, even if up to 48 samples are assayed simultaneously, quartz cuvette chambers and the need for special instrument makes the system limited.

### 2.2. Equipment Demands

In optimal cases, assays should not have any specific demands related to instrumentation. However, for example, CD has its own spectroscopy devices, which are supplied with their own data-processing software packages for data handling. To precisely ascertain the secondary structure of a protein using circular dichroism (CD) data, the collected data must encompass a spectral range that includes wavelengths between 240 and 190 nm [109]. Fortunately, this wavelength range is accommodated by CD instruments [110,111]. TSA is far less equipment-demanding, as in order to collect thermal shift data, only qPCR instruments are recommended. A TSA can be performed using separate PCR and plate reader options if a suitable qPCR is not available [112,113]. However, this is laborious and might cause variation with high sample numbers if the signal is monitored in a well-to-well fashion. Also, the temperature gradient is not constant, potentially affecting the T_m_, but on the other hand, detection sensitivity might be improved due to the higher performance of the used plate reader in comparison to qPCR measurement.

The optimization of protocols for each instrument will depend on its specific software. Researchers should consult the user manual and technical documentation of their particular instrument to understand the available options and recommended protocols for protein thermal denaturation using the TSA [112]. Instruments also offer different usable wavelength ranges and have different sensitivities, partly depending on the use of filters, monochromators, or CCD cameras to detect the signal. Especially in filter-based systems, sensitivity might be significantly reduced if the used filter setting does not fully match the spectra of the used dye. In addition, the excitation can be based on laser, LED, halogen, or xenon lamps, all having varying properties. For example, LightCycler^®^ Real-Time PCR only supports wavelength areas of 430–630 nm, ruling out the suitability for ANS-based TSA assays [114]. Similarly, Anitoa Maverick supports a 460-to-670 nm excitation range and a 510-to-720 nm emission range [115]. On the other hand, the ABI Prism 7000 (Applied Biosystems, Foster City, CA, USA) and Stratagene devices Mx4000 and Mx3000P, for example, can also be used for ANS, as the excitation range is wide (350–750 nm) thanks to the tungsten–halogen lamp used. Related to equipment, a passive reference dye is often used to improve data quality. Usually, ROX (carboxy-X-rhodamine) is used as the passive reference dye when this is needed depending on the qPCR equipment. The main reason to use ROX is to equalize fluorescence levels among wells.

Some instruments might offer built-in templates or preconfigured settings for these assays, simplifying the optimization process [116]. However, if no specific templates are available, researchers may need to customize the protocols based on the instrument’s capabilities and the desired experimental parameters (e.g., temperature range, ramp rate, and data acquisition intervals) [117]. Parameter selection is important to be able to produce comparable data, as these parameters have a known effect on the measured T_m_. Regular calibration and quality control of the real-time PCR instrument is also essential to ensure accurate and reproducible results during protein thermal denaturation experiments [118]. Conducting pilot experiments with known protein standards can also help to fine-tune the protocol and confirm the instrument’s suitability for specific research needs, especially in terms of sensitivity. Overall, proper optimization and understanding of the real-time PCR instrument’s software are crucial for obtaining reliable and informative thermal shift data for protein stability analysis [112].

A promising new variation of DSF that overcomes some of the qPCR-related problems is based on intrinsic fluorescence detection in capillaries (Figure 4). Platforms from NanoTemper Technologies use a capillary-based system in conjunction with a UV detection system designed specifically to monitor shifts in the fluorescence emission of tryptophan by monitoring two wavelengths (330/350 nm) [119]. It can be applied over a wide concentration range (5 μg to 150 mg/mL) and can accommodate for up to 48 samples per run. This device generates high-resolution thermal unfolding curves, enabling the detection and analysis of transitions linked to a multidomain protein [95]. By performing nDSF with Prometheus NT.48, material consumption is significantly lower in comparison to typical DSF, and also, the assay is performed within few minutes in comparison to DSF typically taking an hour. Though powerful, the system costs exceed the budget limit of multiple laboratories. Microscale thermophoresis (MST) is another powerful technique to quantify biomolecular interactions. Although thermophoresis is not a protein thermal stability method, it can be used to determine thermodynamic parameters like ΔG, ΔH, and ΔS by measuring K_d_ at different temperatures. It can also give information about protein–ligand interactions similar to SPR and ITC, making it quite a versatile technique. Thermophoresis shows the directed movement of molecules in a temperature gradient. Size, charge, and conformation along with other molecular properties can be determined based on this movement [120]. Each binding changes at least one these parameters, and as a result, a small volume (~20 μL) of biomolecules is enough for this measurement. This method can be applied to small molecules as well as large macromolecular complexes. A positive aspect is the wide suitability for standard buffers and complex mixtures such as liposomes, detergent, serum, and cell lysates. The Monolith X instrument combines MST with spectral shift, and it is equipped with dual-emission detection optics (NanoTemper Technologies GmbH, München, Germany). This system enables the use of environment-sensitive near-infrared dyes with an emission maximum at around 660 nm, and as with Prometheus NT.48, the measurement step is fast. Also, in this case, the price of the equipment and consumables makes this system suitable for more specialized laboratories or shared facilities.

### 2.3. Buffer Components and Interferences

It has been seen that a loss of function and lower stability occurs for many target proteins in standard sample buffer conditions, and it is essential to identify the components that are necessary to recover the integrity and activity of the protein [121]. The selection of solution conditions for screening by using TSA is heavily reliant on the specific protein being tested and the intended downstream biochemical assays or structural biological approaches [112]. Theoretically, any combination of solution conditions can be tested using the TSA. However, it is essential to consider how different biological buffers may alter the pH with increasing temperature to accurately determine the relationship between pH and protein stability [122,123]. For example, a common problem in DSF experiments is the presence of protein aggregates in the starting sample [46]. The binding and activation of dye by the contaminating aggregate results in high initial fluorescence. Filtering can help, as it removes the interfering signal and the aggregation [46]. The folded protein may be detected by the dye, or it can detect the aggregation in the used assay buffer. To mitigate the impact of this event, adjustments to buffer conditions are sometimes made to stabilize the protein and/or obstruct the dye binding sites [46,124]. In some cases, the assay functionality can be improved by adding co-factors such as Adenosine diphosphate working as endogenous ligand or coordinating metals such as LiCl, or MnCl_2_. Also, optimizing the ionic strength of the buffer and/or including DMSO or sucrose as additives can help to stabilize the protein [46].

TSA can be used to optimize pH/salt in plates to determine the appropriate buffer type, pH, and ionic strength for protein stability studies [125]. The data from different conditions in plates is used to make decisions about these variables, considering a wide range of values while maintaining biological relevance [126]. The profiling is useful in biopharmaceutical protein development and fragment-based drug discovery assays, aiding in identifying optimal buffer conditions for different proteins [127]. Successful formulations also require a balance between protein storage conditions and administration requirements. The matrix of buffer conditions can be applicable to multiple proteins, with individual components further explored to ascertain the ideal buffer condition for a specific protein of interest [128]. pH is the most critical factor for protein formulation because it has a greater effect on protein stability than any other factor, and even other buffer components have an effect on the thermal stability of proteins [123]. One example is the difference in propensity for protein unfolding and aggregation between the sodium acetate and the sodium citrate buffers at the same pH, as reported in stability studies with anti-streptavidin and anti-CD20 monoclonal antibodies [105]. Therefore, both pH and buffer species are key factors to consider when selecting the appropriate buffer for protein formulation [129].

Also, fluorescent dyes are sensitive to their environment, and buffer components might disrupt TSA functionality; thus, the assay must be carefully optimized [87]. pH and presence of detergents and some salts can significantly affect the fluorescent intensity, as observed with fluorescein [130]. Related to typical TSA dyes, it has been shown that EDTA can associate with SYPRO Orange at alkaline pH, thus causing artefacts. Low and high pH are also known to quench its signal [131,132]. Additionally, SYPRO Orange is very sensitive to surfactants and detergents, losing its ability to follow thermal stability changes. Studying protein in buffers containing different concentrations of NaCl, with or without PIPES, showed that different combinations of buffer affect the thermodynamic properties. Stability is also dependent on ionic strength, as was shown with cFMS protein (∆T_m_ = +2.7 °C), by increasing the NaCl concentration from 100 to 500 mM at pH 7 [133]. Also, T_m_ values obtained with buffers of higher molarity were generally lower than those obtained with lower-molarity buffers [134]. In addition to this, some proteins are heavily stabilized in the presence of ions. RASs, for example, carry over magnesium ions from the storage buffer, which might cause changes in stability [26]. In addition to ions, glycerol, which is often used in storage buffers at high concentrations, can increase the protein stability in an assay [135]. Thus, storage buffer conditions have to not only preserve the sample protein, but also to obtain comparable data, especially in case of low-concentration stocks.

### 2.4. Functionality Comparison to Other Methods

The comparison of DSF and microscale DSC (µDSC) to determine the Tm for freeze-dried alpha-1-protease inhibitor (A1PI) showed that comparable results can be obtained, and that both methods show similar trends for the different excipient choices. The results using conventional DSC were also in agreement with these trends, although the measured values were several degrees higher [136]. Sviben et al. confirmed the satisfactory correlation of DSF and DSC, and although DSF is generally less favorable because of the need for a reporter dye, it is superior to DSC with respect to throughput (time per sample in DSC: ∼90 min) [49,134,137]. DSF was compared to ITC; it became evident that DSF has the capability to swiftly and reliably furnish estimates of the dissociation constant for a given protein–ligand combination [138]. In these experiments, the protein consumption was also similar, but the overall assay time with DSF was, again, significantly shorter [138]. In typical DSF assays, protein concentrations of 5–10 µM in a ~20 µL volume are often used [87]. The dyes used are the main limiting factor; for example, SYPRO Orange cannot detect protein at concentration levels below 500 nM, and to obtain a reasonable S/B ratio, usually, a 10-fold-higher protein concentration is used [87]. Using the Protein–Probe family of external peptide-based probes, sensitivity can be lowered down to low nM levels, similar to nDSF. However, these methods cannot reach a similar throughput to SYPRO Orange due to equipment limitations [64].

Traditional DSF has its limitations, including the potential for false positives and false negatives. Assorted methods have been suggested for calculating K_d_ from TSA data, but these approaches still suffer from drawbacks [124]. Moreover, the restricted sensitivity of traditional DSF methods hinders precise K_d_ determination for ligands with affinities falling below the micromolar range.

### 2.5. Data Analysis

Changes in the globular protein due to thermal denaturation can be studied using TSAs. Also, the effect of buffers and small molecules on the protein stability can be investigated using TSAs [112]. Nonetheless, it is essential to highlight that not every folded protein will exhibit ideal profiles during thermal denaturation, complicating the analysis of protein stability. Approximately 15–25% of recombinant proteins may produce non-ideal denaturation curves [139], characterized by high fluorescence at the room temperature baseline and/or a lack of a sigmoidal transition to the unfolded state. Several factors can contribute to non-ideal denaturation profiles [112], including (i) the lack of a compact, globular fold, such as in proteins with intrinsic disorder [140]; (ii) the absence of a hydrophobic core and/or hydrophobic patches on the solvent-exposed surface of the folded protein [141]; and (iii) poor protein stability at room temperature [142]. In cases where these factors are present, the TSA cannot reliably provide information about protein stability or the data are difficult to interpret. Dye selection can help to provide better information, as they have different interaction properties and mechanisms of action, but also, data analysis has an important role [26,113].

The molten globule state is achieved by gradually heating the folded protein. The molten globule is a partially unfolded state of the protein resembling a native-like secondary structure with a loose tertiary structure [89]. The stability of a protein relies on its Gibbs free energy of unfolding, denoted as ∆G_u_, which varies with temperature. In general, as the temperature rises, most proteins become less stable, leading to a decrease in ∆G_u_ until it reaches zero at equilibrium, T_m_, where the concentrations of folded and unfolded proteins are balanced. Melting curve analysis is utilized to ascertain the melting temperature, T_m_ [143,144]. A shift in Tm under various conditions indicates alterations in stability. Equilibrium thermodynamics models apply when the protein unfolds in a reversible two-state manner. When a compound binds to a protein, the free energy contribution from the ligand binding typically increases the ∆G_u_, potentially elevating the T_m_, and this change is called ΔT_m_ [145]. Studies have shown that the stabilizing effect of ligand binding is directly related to the concentration and affinity of the ligands [146]. Although not all proteins undergo reversible (equilibrium) monomolecular two-state reactions during unfolding, assuming equilibrium conditions still allows a reasonable approximation of protein stability, facilitating comparisons under different circumstances [147].

For example, in the case of kinases, the ΔT_m_ of the protein and of the ligand–protein complex has been shown previously to correlate to measures of the ligand’s concentration and binding affinity. In this manner, a melting curve is generated, the T_m_ is determined, and the change (ΔT_m_) induced by prospective binding ligands can be calculated. However, it has been shown that T_m_ shifts do not correlate with protein aggregation propensity; therefore, TSAs cannot be useful regarding information on polymerization [134]. Also, TSAs are not suitable for enveloped viruses, since they are large, biomolecular, multistructured complexes. Although the stabilizing potential of solutions was found to be similar for different proteins, it was found to have different effects on enveloped viruses. It can be concluded that the value of TSAs is great for protein purification and crystallization studies; however, they do not provide information regarding protein polymerization or possible enveloped virus stability [134,148,149].

Depending on the equipment used, there might be a readymade program for data handling in qPCR. In principle, data analysis is fairly simple, but especially for larger sample panels, it can be very time-consuming. Luckily, there are some tools to help with data processing, but most of these tools involve proprietary software and need programming, still only designed for output files from specific instruments. Lately, free and more widely usable software for data handling have been also published [113,150,151]. These methods can easily determine the basic parameters of, e.g., T_m_ and ΔT_m_ based on fitting equations, ranking the stability data, and so on.

## 3. Isothermal Chemical Denaturation (ICD) Assays

Although DSF is heavily used, heating can sometimes cause unwanted effects or not serve as a suitable model to study interactions that need near-physiological conditions [152]. Isothermal chemical denaturation (ICD) is a method used to assess the physical stability of proteins in different formulations either without or with a mild heating step (Figure 5). Thus, ICD is a complementary technique to DSC and DSF based on the thermal denaturation of proteins, and it has been one of the most widely used techniques in academic research laboratories for over 40 years [153,154]. In an ICD experiment, protein samples with increasing concentrations of denaturant (e.g., guanidinium hydrochloride or urea) are prepared, and after incubation, reach equilibrium. Thereafter, denaturation is monitored, often by intrinsic fluorescence, to determine in which denaturant condition the protein unfolding occurs. ICD data are more reliable than DSF regarding the studied protein; however, there remains a need to examine the variation in ΔG across different protein concentrations under identical formulation conditions to evaluate their propensity for aggregation [155]. In denaturation experiments carried out with constant temperatures, it is common for the signal intensity to rise as the protein undergoes unfolding, similar to the signal increase in DSF upon temperature increase [156]. Various factors can affect the observed signal intensity in ICD measurements; for example, for sufficiently dilute samples, the signal increases linearly with the protein concentration, but also, solvent conditions can affect the intensity by changing the electrostatic environment of exposed tryptophan (Figure 5) [156]. Within a typical biopharmaceutical pH range, the pH can have a small, nonlinear effect on fluorescence intensity, and also, protein aggregation is known to decrease the observed signal [156,157].

In principle, both thermal and chemical denaturation techniques may be used to determine ΔG as a measure of a protein’s intrinsic conformational stability. However, in practice, a limitation of thermal unfolding techniques can be the irreversibility of denaturation due to aggregation and precipitation invalidating thermodynamic analysis [95]. Formulation studies may be significantly aided by using chemical denaturation as a reliable technique for measuring ΔG and its response to pH, ionic strength, excipients, and protein concentration. ICD consumes a higher volume of samples and has a lower throughput than DSF, which make it less favorable in biopharmaceutical screening settings [156]. A cutting-edge HTS instrument based on ICD has recently been developed. This instrument is capable of automatically preparing and measuring protein samples that encompass a range of denaturant concentrations [153].

DSF and thermal denaturation techniques in general can give inaccurate insights into formulations due to several reasons, such as changes in pH during heating. Therefore, it is preferred to use both ICD and DSF in combination to evaluate the physical stability of protein in formulations. Investigations with mAbs have shown that the concentration dependence of ΔG_u_ determined by ICD could be used for understanding the physical stability of a protein in different formulations along with DSF [158]. One other drawback of ICD is the long incubation period, and also that measurements are taken one by one at specific denaturant concentrations [159]. Nonetheless, this enables access to the complete emission spectra, unlike in DSF, providing the opportunity to employ advanced data analysis methods. This allows for (a) a more in-depth examination of the fluorescence peak shift, (b) a more precise characterization of the shift and unfolding ratio, and (c) an enhanced ability to differentiate between various unfolding transitions [156]. One way of analyzing the data from ICD experiments is to measure the ΔG values of samples by collecting and analyzing 32-point denaturation curves on a HUNKY system (Unchained Labs, Pleasanton, CA, USA) [153]. As an example, each sample can be denatured with a linear gradient of urea from 1.5 to 6.5 M at 25 °C by incubating samples for 3 h prior to fluorescence data acquisition (F350/330). Denaturation curves are thereafter analyzed with the HUNKY analysis software (https://www.unchainedlabs.com/), using the barycentric mean (BCM) fluorescence analysis method and a three-state fitting model [153].

### 3.1. Fluorescent Dyes vs. Intrinsic Fluorescence

External fluorescent probes are used more frequently in DSF assays than intrinsic fluorescence, but in ICD, the situation turns the other way around. In intrinsic ICD, protein unfolding is characterized by measuring the ratio between the exposed solvent and the buried tryptophan fluorescence signal peak shapes equal to DSF (Figure 5A) [156]. These residues can be excited at wavelengths in the range of 260–280 nm and with emission spectra starting from 320 nm. On a per residue basis, the tryptophan fluorescence is the predominant contributor to the total intrinsic fluorescence, and usually, a 330-to-350 nm ratio is used for the final results due to the expected spectral red shift. As an example, mAbs typically contain approximately 20 tryptophan residues, and thus, they can be efficiently studied using intrinsic fluorescence. However, some proteins might have no tryptophan moiety, which severely limits the use and sensitivity of intrinsic fluorescence and might prevent measurements completely.

In some special scenarios, intrinsic FRET (iFRET), a technique which utilizes tryptophan residues of the target proteins and an added target-specific acceptor probe, could be a usable option, but this cannot overcome the problem if the assay functionality is weak due to the lack of tryptophan. Thus, there has been attempts to use DSF dyes in ICD experiments. Different fluorescent dyes have been tested for ICD, such as 1,8-ANS, Dapoxylbutyl sulfonamide (DBS), Nano Orange, SYPRO Orange, and SYPRO red. Based on these results, Nano Orange fluorescence provided a viable option for model proteins thymidylate kinase and stromelysin. This was because it had the lowest background fluorescence of any of the dyes listed [160]. It is also observed that adding dye to the denatured protein results in an instantaneous increase in the fluorescence. Using this sharp peak, kinetic studies are feasible [160]. Unfortunately, the denaturant effect on DSF dyes reduces their usability. It is known that pH, used also as a denaturant, affects, e.g., SYPRO Orange signals [161]. Using bovine serum albumin (BSA) it has also been shown that the ANS fluorescence lifetime is reduced in the presence of guanidine hydrochloride (GdnHCl), a second, often used denaturant. Similar GdnHCl-induced effects are also reported with Nile Red and human serum albumin (HSA). In this scenario, the fluorescence intensity peaked between 0.25 and 1.5 M of GdnHCl, subsequently decreasing beyond 1.5 M of GdnHCl. This decrease was accompanied by a red shift in the emission maximum, transitioning from 620 to 645 nm [100]. These effects at high denaturant concentrations not only complicate assay result interpretation and cause uncertainty, but also might avoid assays with the most stable proteins.

An interesting approach to combine good properties of DSF and ICD was recently introduced by using a FRET-Probe [162]. This assay is suitable mainly for PLI studies, and unlike typical ICD, it is performed in a single optimized denaturant concentration following the denaturation process over time. It was shown that pH, alcohols, and urea can be used as denaturants, and binding information for even mid-nM affinity binders can be easily obtained. However, the FRET-Probe seems not to tolerate harsh denaturant conditions, which limits the assay use for the most stable proteins. The FRET-Probe was shown to also enable DSF-type monitoring using temperature for denaturation, but this approach is limited due to a lack of qPCR enabling TR-FRET signal monitoring.

### 3.2. Denaturants and Their Function

Unsurprisingly, denaturant selection is major step in ICD assay optimization. Different denaturants have different mechanism of action, and thus, some proteins are more tolerant to one denaturant than to others. Guanidine salts, especially GdnHCl and guanidine thiocyanate, are commonly used denaturants, as well as urea, which is a nonionic and milder alternative for denaturation [163,164]. These denaturants can also be used as ultra-strong thiourea, but also, milder options like alcohols and pH have been used [165,166,167]. In the presence of chemical denaturants such as urea and GdnHCl, the Gibbs energy has been experimentally shown to follow a simple linear dependence on the denaturant concentration [168]. A positive ΔG indicates that the native state is more stable than the denatured state. The denaturation midpoint corresponds to the denaturant concentration at which ΔG equals zero. Chemical denaturation experiments provide three key parameters: ΔG°, the m value, and C_m_. Among these, C_m_ stands out as the parameter that can be determined with the least experimental error [169]. Chemical denaturation in general and urea denaturation in particular allow for the identification of solvent conditions that maximize the structural stability of a protein. Shifting the denaturation curves for higher C_m_ values, higher urea concentrations, or higher ΔG° values mean increasing protein stability, which can be achieved with several parameters such as pH, salts, ligands, and excipients [154]. The denaturation process initiated by urea involves essential steps: the solvation of the protein backbone through hydrogen bonding, a predilection for electrostatic interaction with hydrophilic residues, and a dispersion interaction with hydrophobic residues. These interactions collectively contribute to the intrusion of urea into the protein core and subsequent denaturation [170]. Using human placental cystatin as an example, C_m_ is observed at 1.5 M of GdnHCl or 3 M of urea, and it greatly loses its structure at 6 M of urea, completely forming a random coil structure at an 8 M concentration [171]. As another example, in a urea assay with carbonic anhydrase (CA) and its inhibitor trifluoromethanesulfonamide (TFMSA), the curve shifted to higher urea concentrations consistent with the stabilization of the CA. Chemical denaturation experiments, especially urea denaturation, which does not affect ionic strength as GdnHCl does, provide a way to decouple temperature from solvent effects in irreversible denaturation [154]. Proteins susceptible to irreversible denaturation at elevated temperatures often exhibit a propensity for undergoing reversible denaturation when exposed to urea at lower temperatures [172].

Urea causes protein unfolding, either directly through interactions with protein hydrophobic parts and water molecules, or indirectly through alterations to the solvent composition [170,173,174]. Alcohols such as ethanol, methanol, propanol, and butanol have been proposed for the chemical denaturation of proteins due to their high content of hydrocarbon and water miscibility. These properties facilitate the unfolding of native structures at a relatively low dose [162]. Less used denaturants in the context of ICD are pH and SDS. SDS is often used as a denaturant in gel electrophoresis, but its chemical nature as a surfactant makes SDS difficult for ICD due to the formation of micelles. The buffer pH is known to have a significant effect on protein stability, which needs to be additionally considered in all assay designs [162]. In specific cases, investigations at various pH levels can give valuable information, as shown, for example, with cytochrome c (cyt c). It was shown that the C_m_ increases at higher pH values, but also that cyt c can recover its native structure after exposure to extremely low pH [154,175]. The use of pH can also be combined with other denaturants like urea and GdnHCl, as shown with cyt c [176]. Sodium dodecyl sulfate (SDS), an anionic surfactant that is commonly used to mimic hydrophobic binding environments such as cell membranes, is known to denature some native-state proteins, including cyt c [177]. SDS denatures proteins by forming protein-decorated micelles and, similar to urea, it breaks non-covalent interactions [178]. Sodium sulfite and sodium hydrogen sulfite (SHS) induce denaturation by breaking disulfide bonds, and the reduction in chemical crosslinking exposes the protein to denaturation. These and other denaturants are, however, not widely used.

### 3.3. ICD Comparison to TSA

When ICD and TSA are compared, ICD is mainly used in research when TSA is not a suitable option [179]. Especially in case of ligand binding studies, TSA is more often used. ICD does not offer the same throughput capabilities as TSA, but it provides more precise information about protein stability and interactions, and thus, in cases of formulation, it is often a valuable option. ICD has a limited sensitivity and micromolar range which is similar to DSF-type assays. However, whilst DSF is performed in a single condition, ICD measurements are made by titrating the denaturant using varying concentrations for a single experiment, increasing the protein consumption significantly [162]. It has been suggested that a combination of DSF and ICD would be feasible to reduce the protein amount required to assess the physical stability in various formulations but still provide a sufficient prediction quality [180]. Indeed, assays performed with the FRET-Probe using a single denaturant concentration supports this idea [162]. A comparison of the sensitivity of DSF and ICD in detecting the ligand binding of reduced nicotinamide adenine dinucleotide (NADH) with malate dehydrogenase not only validates ICD as a reliable method for screening for ligands, but also indicates that, in some cases, ICD is more sensitive than DSF at detecting binding at lower concentrations of ligands [181]. This same interaction was also studied with the FRET-Probe, showing superior performance in comparison to both ICD and DSF.

ICD also serves as an alternative and complementary technique to DSC and may circumvent common issues that can arise with thermal unfolding (e.g., precipitation). With the advent of new, automated tools that dramatically increase the throughput of chemical denaturation studies, it is critical to pay special attention to the experimental details concerning sample preparation, measurements, and data analysis in order to achieve accurate, reproducible data via ICD [159]. Without performing CD or other control experiments, even qualitative rankings of protein stability using ICD data alone may be incorrect.

As DSF is typically used for globular and soluble proteins, ICD expands the available toolkit of biophysical techniques to characterize and study ligand binding to integral membrane proteins. In the assessment of weak compound affinity through ICD, the limitation often stems from their solubility. As a practical guideline, achieving optimal results entails maintaining a ligand concentration at least two orders of magnitude higher than the K_d_ [182]. mAb1 in different buffers with ICD and DSF showed that the C_m_ values of mAb1 in histidine are similar to or higher than the C_m_ values in citrate or phosphate formulations with the same pH, while the Tm values of mAb1 in histidine formulations were lower compared to their citrate and phosphate counterparts. One reason for this is that ICD is an isothermal technique, and any pH or temperature drift of excipients is avoided [180].

## 4. Future Directions

The market of protein stability analysis is growing rapidly and it is already worth approximately USD 2 billion, from which reagents and assay kits have the largest share. Pharmaceutical and biotechnology companies are the main players in this field, especially due to their drug discovery applications. But these assays are also very often used in academic and research institutes, though the exact methods utilized are slightly different, mainly due to regulations and assay scale. Although protein stability assays have been performed for decades, new ways to utilize the data and new and improved methods and instruments are still being constantly developed. Protein folding and stability are complex processes related to multiple aspects of the protein environment, and we still have to gain new information to learn how to reliably predict their stability [183]. Until that, stability assays will hold their place as valuable research and screening tools.

Depending on the stability assay technique, the throughput and structural resolution might significantly vary, as well as whether the stability is directly or indirectly measured [184]. Also, the precision and accuracy of measurements might vary depending on each technology, but also the need. Technologies are developing in multiple directions and their application number is increasing. Positive aspects of direct, small-scale and indirect, large-scale measurements of protein stability have been one of the main focuses. This relates to liquid handling automation and miniaturization, which can increase the throughput [184]. On the other hand, mapping the effects collected from large screening campaigns, like mass spectrometry-based thermal proteome profiling (TPP), can indirectly produce stability data [19,20]. This can be further helped by novel machine learning architectures, at least to first classify proteins as thermostable and thermolabile, and later to even predict more exact properties, like the protein T_m_ [183,185].

Behind these major trends, many researchers, especially in academic and research institutes, will still rely on DSF- and ICD-type simple, cheap, and non-equipment-limited assays in their research. DSF is a potent tool, especially in PLI studies, to validate and optimize a ligand and its interaction [26]. However, it is quite prone to errors and often can only give yes/no answers [180]. To improve the method, new external probes and labeled-protein-based methods have been introduced to increase not only the robustness but also the sensitivity [2,64,162]. Increased sensitivity can improve the DSF resolution from the protein to the domain level, and in addition, improve the collection of true binding affinity information with high-affinity ligands [64,162]. In addition, nDSF obviates the need for dyes, allowing for membrane protein studies, as typically, DSF is used only for soluble proteins. However, nDSF’s availability is currently still limited. DSF can also be used for optimizing buffer ingredients, but ICD and combinatory methods are often more reliable tools to predict the physical stability of a model protein [162,180]. Formulation relates to protein stability along with aggregation properties. These factors are becoming more important day by day as biologics become more common [186,187,188]. The advantage of isothermal approaches is that proteins can be studied in more physiological conditions, reducing the potential negative effects caused by heating the buffer components [123,180]. Thus, future external probes with high sensitivity and functionality in both DSF and ICD settings could bring reliable and easy-to-use systems available for all [162].

## Figures and Tables

**Figure 1 ijms-25-01764-f001:**
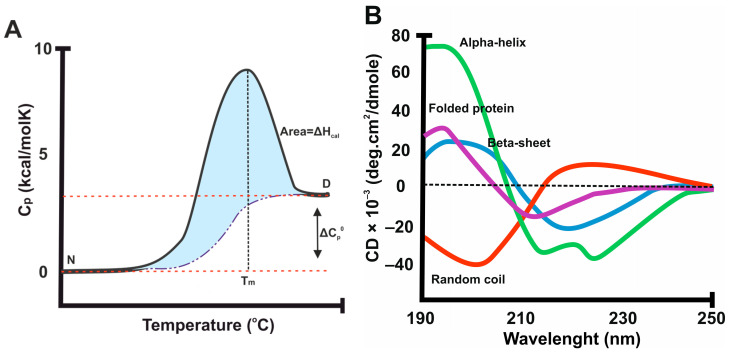
Typical data obtained using differential scanning calorimetry (DSC) and circular dichroism (CD). (**A**) Typical DSC thermogram of a small globular protein provides T_m_ at maximum when the heat capacity (C_p_) changes during the temperature increase, causing native (N) protein unfolding to denatured (D) form. (**B**) Typical circular dichroism (CD) spectra of proteins’ most important secondary structures, and comparison between folded and unfolded protein spectra.

**Figure 2 ijms-25-01764-f002:**
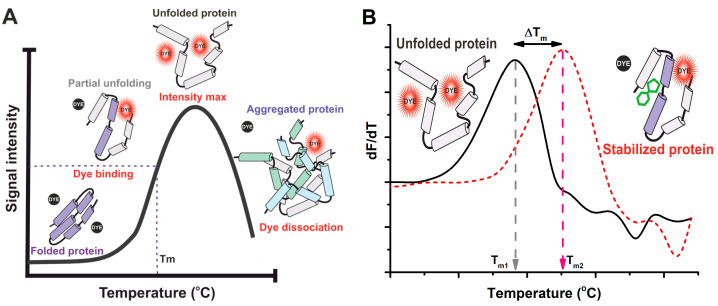
Typical thermal stability assay (TSA) data using external fluorescent dye. (**A**) Increase in temperature causes protein unfolding, enabling dye binding and increase in fluorescence. Melting temperature (T_m_) is observed as a half-maximum signal, and further increase in temperature leads to dye dissociation due to protein aggregation. (**B**) Data can be additionally plotted as a slope of the fluorescence curve (dF/dT) vs. temperature to improve visualization. T_m_ is shown as a peak and ΔT_m_ can be obtained as a distance between the peaks of a ligand-bound (red) and non-bound (black) protein.

**Figure 3 ijms-25-01764-f003:**
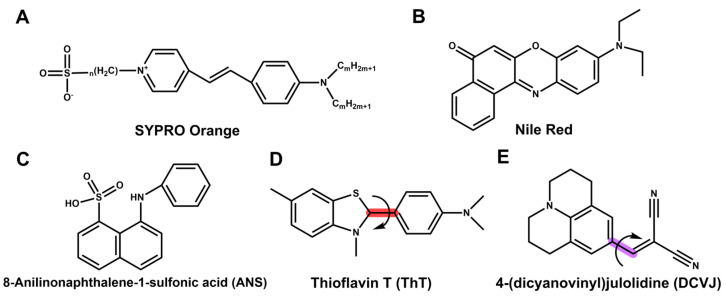
Typical external fluorescence dye structures used to study proteins. (**A**) SYPRO Orange is typical and most often used DSF dye; Ex: 470 nm; Em: 570 nm. (**B**) Nile Red is a polarity-sensitive fluorescent probe; Ex: 550 nm; Em: 635 nm. (**C**) ANS is a fluorescent molecular probe binding to hydrophobic regions of target protein undergoing fluorescence blue-shift upon binding; Ex: 350 nm (free); Em: 520 nm (free). (**D**) ThT dye is used to visualize and quantify the presence of misfolded protein aggregates; Ex: 450 nm; Em: 485 nm. (**E**) DCVJ is a rotor dye, with which quantum yield increases by decreasing free rotation upon binding; Ex: 450 nm; Em: 500 nm.

**Figure 4 ijms-25-01764-f004:**
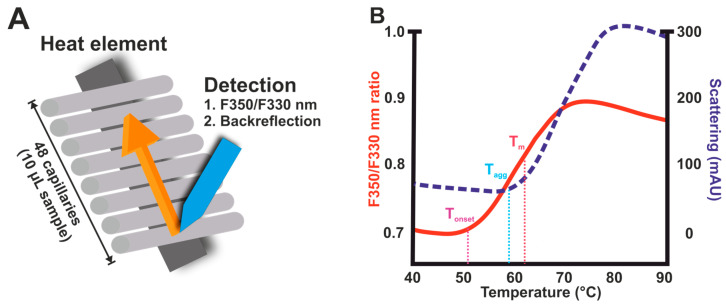
NanoDSF (nDSF) principle and typical data observed. (**A**) nDSF assays are performed in small capillaries and the assay enables the detection of conformational stability, fluorescence scan using 330 and 350 nm wavelengths, and colloidal stability by monitoring back-reflection light scattering simultaneously. (**B**) nDSF fluorescence measurement at two wavelengths, 330 and 350 nm, provides a typical TSA melting curve based on the changes in the tryptophan environment. Aggregation (blue) typically occurs at higher temperatures in comparison to unfolding (red) and reliable data are only observed at higher concentrations, approximately 20× above the DSF sensitivity limit.

**Figure 5 ijms-25-01764-f005:**
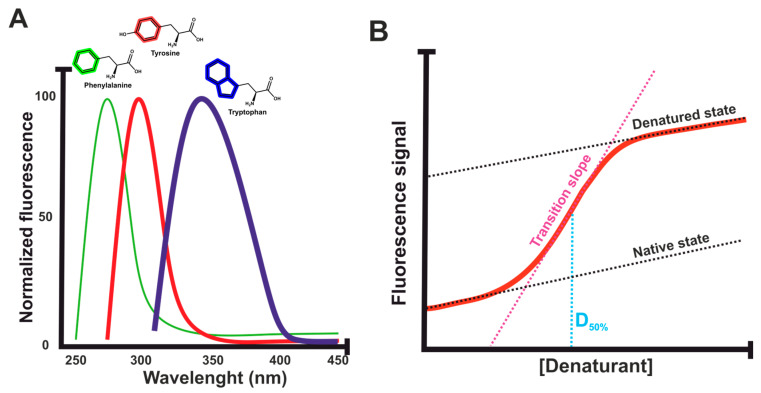
Protein intrinsic fluorescence and isothermal chemical denaturation (ICD). (**A**) The use of intrinsic fluorescence detection is the most powerful for proteins with tryptophan moieties with the best fluorescence properties, followed by tyrosine and phenylalanine, which limit not only the quantum yield, but also low-wavelength emission and excitation. Red shift in tryptophan fluorescence typically occurs during protein denaturation and its fluorescence is also affected by nearby tyrosine. (**B**) ICD is performed in a selected temperature, typically RT or 37 °C, performing denaturant titration with each sample and monitoring the signal at equilibrium. Often, chemical denaturation is reversible, unlike temperature-induced unfolding, which in many cases is irreversible.

## Data Availability

Not applicable.

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
