# Peer review of "Fluorescence-Based Protein Stability Monitoring—A Review"

_ijms, 2024, doi:10.3390/ijms25031764_

Round 1

Reviewer 1 Report

Comments and Suggestions for Authors

In their review of fluorescence-based protein stability measurements Gooran and Kopra provide an extensive overview of the application of this biophysical measurement in analytical protein science studies. They discuss the suitability of DSF measurements for determining protein stability and how these assays can be impactful in buffer optimization, formulations studies and crystallization efforts. They also highlight how DSF is implemented in characterizing protein:ligand interactions. Their inclusion of isothermal chemical denaturation is a nice complement to the temperature induced stability studies they describe in the DSF measurements. This reviewer believes that this article will be a valued addition to the literature describing measurements of protein stability and undoubtably be a useful resource for identifying publications that successfully implemented protein stability measurements in protein science studies.  There are no major revisions required, but some minor revisions needed to clarify some points, or to improve the language.

Line 12: replace “but main focus…” with “but the main focus….”

Line 30: remove e.g., replace with “and what for example is the isoelectric point (pI)”

Line 31: replace “by means of e.g. pH, ions, and temperature.” with “by defining optimal pH, ions and temperature”

Line 46: remove e.g. – the sentence reads perfectly without it.

Line 91:  “method of high importance” should read, “method is of high importance”

Line 92: “Research that grants mass measurement of molecules simultaneously with the possibility of repetition is called high throughput.” – the use of mass measurements is confusing, perhaps replace with multiple?

Line 96: “They usually are easy..” change to “They are usually easy..”

Line 104: “disturbing and distressing the protein inadequately and measure the stability precisely.” This phrase is unclear what exactly is meant. Suggest re-writing it for clarity.

Line 114: replace “scientist” with “scientists”

Line 127: “When ΔG paired with Tm” replace with “When ΔG is paired with Tm”

Line 156: “However, not always this high resolution is needed, but domain or even protein level resolution might give the needed information.” The meaning of this sentence is unclear, suggest rewriting it.

Line 195: “A fluorescence dye, instead of detecting the presence of amplified nucleotides, is used to detect binding associated with the appearance of solvent-exposed hydrophobic regions after unfolding.” Replace with “Instead of detecting the presence of amplified nucleotides, a fluorescence dye is used to detect binding associated with the appearance of solvent-exposed hydrophobic regions after unfolding”

Line 231: “CD is a long-established method..” suggest replacing with “CD is a well-established method..”

Line 257: “Ideal assay for universal..” replace with An ideal assay for universal…”

Line 286: “The exploration of the small-molecule correctors…” Unclear what small molecule correctors are? Rephase or provide more details.

Line 301: “blotted” should read “plotted”

Line 384: In the discussion of the use of nanoDSF the authors state “Unfortunately, these as-384 says are still out of reach for many scientists.” I assume this is due to the cost of specialized equipment, however the authors should state that or make it clearer why these methods are out of reach of most scientists.

Line 405: “Used near-infrared dyes..” – I think this should read “The use of near-infrared dyes..”

Line 407: decree should read “degree”

Line 450: “Usually, ROX is used as this type of reference, but depending on the equipment used for reference might not be needed.” The meaning of this sentence is not clear – suggest it be re-written. Also, what is ROX? – please specify.

Line 480: “By performing nDSF with Prometheus NT.48, material consumption is significantly lower in comparison to 481 typical DSF, and also assay is performed while waiting.” It is not clear the significance of the bolded statement – aren’t DSF assays using RT-PCR instruments “performed while waiting”. Suggest re-writing the sentence for clarity.

Line 484: “Thermophoresis is the base of this method…”, base should be replaced with “basis”. The inclusion of MST doesn’t quite fit with the overall description of fluorescence-based protein stability – this method is focused on measuring protein interactions and does not rely on protein unfolding for the measurement. Suggest taking the discussion of MST out.

Line 515; “Also, accustoming the ionic strength of the buffer”, do the authors mean accustoming, as the definition of this word is accepting something as normal or usual. Would optimizing be an alternative word to use?

Line 537: “fluorescent intensity like shown e.g. with fluorescein” – this is not grammatically correct, change to “fluorescent intensity as observed with fluorescein”

Line 548: “Like shown with RAS, even carryover ion (Mg2+) from the storage buffer might cause stability changes” suggest changing to, “RAS, for example, carry over magnesium ions from the storage buffer might cause changes in stability”

Line 566-567:  “Used dyes are the main limited factor, as dyes like SYPRO Orange” suggest replacing with “The dyes used are the main limiting factor, for example SYPRO Orange…”

Line 622 “Depending on the used equipment, there might be a readymade program for data handling in the qPCR.” Replace with “Depending on the equipment used, there might be a readymade program for data handling in the qPCR.”

Line 631: “Even DSF is heavily used…” change to “Although DSF is heavily used..”

Line 639: “after incubation to reach equilibrium” replace with “after incubation reach equilibrium”

Line 646: “Various factors can affect the observed signal intensity ICD,” this part of the sentence doesn’t make sense – maybe missing a word? Perhaps should read “Various factors can affect the observed signal intensity in ICD measurements,”

Line 658: “ICD consumes higher volume of samples and have lower..” replace have with has.

Line 688: “Denaturation curves is thereafter” replace is with are

Line 692: “In DSF assays, the use of external fluorescent probes is more often as intrinsic fluorescence, but is ICD the situation turns other way around” – this isn’t clear, do the authors mean “External fluorescent probes are used more frequently in DSF assays than intrinsic fluorescence, but in ICD the situation turns other way around” ??

Line 699: “Due to the demand of tryptophan, mAb’s, typically contains approximately 20 tryptophan residues, can often be efficiently studied.” This sentence is not clear and should be re-written.

Line 704: “could be usable option”, replace with “could be a usable option”

Line 709: “for model protein” replace with “for model proteins”

Line 711: “It is also observed that when the protein was first denatured and the dye was added at the end of the process, the fluorescence increase was instantaneous, even most of the studies have been performed in a kinetic fashion” The meaning of this sentence is not clear, suggest rewriting.

Line 724: “Interesting approach to” replace with “An interesting approach to”

Line 736: “one denaturant than to other” replace with “one denaturant than to others

Line 765: “As urea causes protein unfolding” remove “As” to read “Urea causes protein unfolding”

Line 770: “Less used denaturants in a context of ICD and pH and denaturants as SDS, often used in gel electrophoresis as a denaturant.” The meaning of the sentence is not clear, suggest rewriting.

Line 776; “The use of pH can also be combined with other denaturants like urea and GdnHCl like also shown with cyt c.” replace with “The use of pH can also be combined with other denaturants like urea and GdnHCl as shown with cyt c.”

Line 786: “When ICD is and TSA is compared” replace with “When ICD and TSA are compared”

Line 788: “ICD do not” replace with “ICD does not”

Line 790: “ICD also suffers from limited sensitivity, which typically in micromolar range like with DSF-type assays, but due to titrations with varying denaturant concentrations the protein consumption is significantly higher” This sentence is unclear, suggest rewriting.

Line 831: “Until that, stability assays will hold their place as and valuable research and screening tools.” Remove “and”, replace with “Until that, stability assays will hold their place as valuable research and screening tools.”

Line 837: “Positive aspects related to direct but small-scale and large scale but inferential measurements of protein stability have been one of the main focuses. “ Rewite, meaning not clear.

Line 857: “Formulation relates to protein stability, but also aggregation properties, and these factors are becoming all the time more important, as biologics have become more common” sentence poorly written, suggest rewriting.

Comments on the Quality of English Language

There are several parts of the manuscript where the meaning is unclear - this is most likely due to the authors not being native English speakers. Suggested edits to the language have been made in the previous section

Author Response

In their review of fluorescence-based protein stability measurements Gooran and Kopra provide an extensive overview of the application of this biophysical measurement in analytical protein science studies. They discuss the suitability of DSF measurements for determining protein stability and how these assays can be impactful in buffer optimization, formulations studies and crystallization efforts. They also highlight how DSF is implemented in characterizing protein:ligand interactions. Their inclusion of isothermal chemical denaturation is a nice complement to the temperature induced stability studies they describe in the DSF measurements. This reviewer believes that this article will be a valued addition to the literature describing measurements of protein stability and undoubtably be a useful resource for identifying publications that successfully implemented protein stability measurements in protein science studies.  There are no major revisions required, but some minor revisions needed to clarify some points, or to improve the language.

We sincerely thank the Reviewer for positive evaluation of our manuscript and for offering very helpful comments to help us improve the manuscript. We fully addressed the comments made by the Reviewer as described point-by-point below

Line 12: replace “but main focus…” with “but the main focus….”

Line 30: remove e.g., replace with “and what for example is the isoelectric point (pI)”

Line 31: replace “by means of e.g. pH, ions, and temperature.” with “by defining optimal pH, ions and temperature”

Line 46: remove e.g. – the sentence reads perfectly without it.

Line 91:  “method of high importance” should read, “method is of high importance”

Line 92: “Research that grants mass measurement of molecules simultaneously with the possibility of repetition is called high throughput.” – the use of mass measurements is confusing, perhaps replace with multiple?

Line 96: “They usually are easy..” change to “They are usually easy..”

Line 114: replace “scientist” with “scientists”

Line 127: “When ΔG paired with Tm” replace with “When ΔG is paired with Tm”

Line 195: “A fluorescence dye, instead of detecting the presence of amplified nucleotides, is used to detect binding associated with the appearance of solvent-exposed hydrophobic regions after unfolding.” Replace with “Instead of detecting the presence of amplified nucleotides, a fluorescence dye is used to detect binding associated with the appearance of solvent-exposed hydrophobic regions after unfolding”

Line 231: “CD is a long-established method..” suggest replacing with “CD is a well-established method..”

Line 257: “Ideal assay for universal..” replace with An ideal assay for universal…”

Line 301: “blotted” should read “plotted”

Line 405: “Used near-infrared dyes..” – I think this should read “The use of near-infrared dyes..”

Line 407: decree should read “degree”

Line 537: “fluorescent intensity like shown e.g. with fluorescein” – this is not grammatically correct, change to “fluorescent intensity as observed with fluorescein”

Line 548: “Like shown with RAS, even carryover ion (Mg2+) from the storage buffer might cause stability changes” suggest changing to, “RAS, for example, carry over magnesium ions from the storage buffer might cause changes in stability”

Line 566-567:  “Used dyes are the main limited factor, as dyes like SYPRO Orange” suggest replacing with “The dyes used are the main limiting factor, for example SYPRO Orange…”

Line 622 “Depending on the used equipment, there might be a readymade program for data handling in the qPCR.” Replace with “Depending on the equipment used, there might be a readymade program for data handling in the qPCR.”

Line 631: “Even DSF is heavily used…” change to “Although DSF is heavily used..”

Line 639: “after incubation to reach equilibrium” replace with “after incubation reach equilibrium”

Line 646: “Various factors can affect the observed signal intensity ICD,” this part of the sentence doesn’t make sense – maybe missing a word? Perhaps should read “Various factors can affect the observed signal intensity in ICD measurements,”

Line 658: “ICD consumes higher volume of samples and have lower..” replace have with has.

Line 688: “Denaturation curves is thereafter” replace is with are

Line 704: “could be usable option”, replace with “could be a usable option”

Line 709: “for model protein” replace with “for model proteins”

Line 724: “Interesting approach to” replace with “An interesting approach to”

Line 736: “one denaturant than to other” replace with “one denaturant than to others”

Line 765: “As urea causes protein unfolding” remove “As” to read “Urea causes protein unfolding”

Line 776; “The use of pH can also be combined with other denaturants like urea and GdnHCl like also shown with cyt c.” replace with “The use of pH can also be combined with other denaturants like urea and GdnHCl as shown with cyt c.”

Line 786: “When ICD is and TSA is compared” replace with “When ICD and TSA are compared”

Line 788: “ICD do not” replace with “ICD does not”

Line 831: “Until that, stability assays will hold their place as and valuable research and screening tools.” Remove “and”, replace with “Until that, stability assays will hold their place as valuable research and screening tools.”

All the above suggestions have been corrected to the manuscript and we have tried to improve clarity and language in all steps.

Line 104: “disturbing and distressing the protein inadequately and measure the stability precisely.” This phrase is unclear what exactly is meant. Suggest re-writing it for clarity.

The sentence has been changed to: incomplete distressing and disturbing of the protein and measure the stability accurately

Line 156: “However, not always this high resolution is needed, but domain or even protein level resolution might give the needed information.” The meaning of this sentence is unclear, suggest rewriting it.

The sentence has been changed to: However, the high resolution is not always needed, as protein level resolution might give the needed information

Line 286: “The exploration of the small-molecule correctors…” Unclear what small molecule correctors are? Rephase or provide more details.

Corrector are stabilizing molecules which stabilize mutant protein to a wild-type-like conformation. We have added the explanation to the text.

Line 384: In the discussion of the use of nanoDSF the authors state “Unfortunately, these as-384 says are still out of reach for many scientists.” I assume this is due to the cost of specialized equipment, however the authors should state that or make it clearer why these methods are out of reach of most scientists.

This is a good comment and we have been unclear. Costs for a single experiment is the main thing of course, but also that the equipment’s are not available in all research facilities, also a cost issue. We have clarified this statement in the text

Line 450: “Usually, ROX is used as this type of reference, but depending on the equipment used for reference might not be needed.” The meaning of this sentence is not clear – suggest it be re-written. Also, what is ROX? – please specify.

The sentence is changed and full name of ROX is added. ROX is an passive dye not affected by the reaction studied, but can be used as to estimate potential interferences and correct well-to-well fluorescence variation.

Line 480: “By performing nDSF with Prometheus NT.48, material consumption is significantly lower in comparison to 481 typical DSF, and also assay is performed while waiting.” It is not clear the significance of the bolded statement – aren’t DSF assays using RT-PCR instruments “performed while waiting”. Suggest re-writing the sentence for clarity.

We have now clarified this statement and we agree that it was unclear. Both assays are quite fast, but the time scale minutes in nDSF in comparison to about an hour with qPCR is still different. Also the applicability and the type of experiments performed is different, as nDSF is not really screening applicable, but more efficient with small number of molecules and to estimate their binding affinity.

Line 484: “Thermophoresis is the base of this method…”, base should be replaced with “basis”. The inclusion of MST doesn’t quite fit with the overall description of fluorescence-based protein stability – this method is focused on measuring protein interactions and does not rely on protein unfolding for the measurement. Suggest taking the discussion of MST out.

The reason why thermophoresis is shortly discussed here is Nanotemper Monolith X, but we have been unclear and it is through that it is not directly related to protein stability but screening. Monolith X has both the MST and thermal-shift properties, as it is kind of option for SPR and ITC. However, it can also be used to determine thermodynamical parameter by monitoring Kd in different temperatures and thus we wanted to include it. We have modified this section to clarify our point and removed some of the discussion.

Line 515; “Also, accustoming the ionic strength of the buffer”, do the authors mean accustoming, as the definition of this word is accepting something as normal or usual. Would optimizing be an alternative word to use?

Yes, “accustoming” is changed to “optimizing”

Line 692: “In DSF assays, the use of external fluorescent probes is more often as intrinsic fluorescence, but is ICD the situation turns other way around” – this isn’t clear, do the authors mean “External fluorescent probes are used more frequently in DSF assays than intrinsic fluorescence, but in ICD the situation turns other way around” ??

Yes, the sentence is changed to the suggested one.

Line 699: “Due to the demand of tryptophan, mAb’s, typically contains approximately 20 tryptophan residues, can often be efficiently studied.” This sentence is not clear and should be re-written.

The sentence is changed and we have clarified our point. The number was given just to give an idea why antibodies are highly visible and suitable for this type of assays. Not all proteins contain tryptophan’s.

Line 711: “It is also observed that when the protein was first denatured and the dye was added at the end of the process, the fluorescence increase was instantaneous, even most of the studies have been performed in a kinetic fashion” The meaning of this sentence is not clear, suggest rewriting.

The sentence is changed to: “It is also observed that adding dye to the denatured protein results in instantaneous in-crease of the fluorescence. Using this sharp peak, the kinetic studies are feasible”

Line 770: “Less used denaturants in a context of ICD and pH and denaturants as SDS, often used in gel electrophoresis as a denaturant.” The meaning of the sentence is not clear, suggest rewriting.

The sentence corrected and to make it more clear, it is divided in two sentences. Additionally we added explanation why SDS is more rarely used.

Line 790: “ICD also suffers from limited sensitivity, which typically in micromolar range like with DSF-type assays, but due to titrations with varying denaturant concentrations the protein consumption is significantly higher” This sentence is unclear, suggest rewriting.

The sentence was rephrased to clarify why protein consumption is larger than in DSF even the sensitivity is similar.

Line 837: “Positive aspects related to direct but small-scale and large scale but inferential measurements of protein stability have been one of the main focuses. “ Rewite, meaning not clear.

The sentence is changed to: “Positive aspects of direct-small-scale and indirect-large-scale measurements of protein stability have been one of the main focuses.”

Line 857: “Formulation relates to protein stability, but also aggregation properties, and these factors are becoming all the time more important, as biologics have become more common” sentence poorly written, suggest rewriting.

The sentence is changed to: “Formulation relates to protein stability along with aggregation properties. These factors are becoming more important day by day, as biologics have become more common”

Reviewer 2 Report

Comments and Suggestions for Authors

General comment:

This manuscript, entitled “Fluorescence-based protein stability monitoring – a review,” by Gooran and Kopra, reviewed the fluorescence-based low instrument and expertise demand techniques to study protein stability related to structure and function. Also, they compared differential scanning fluorimetry (DSF) to isothermal chemical denaturation (ICD). A descriptive survey like this helps to find protein stability in a different environment to compare with its counterpart by creating mutation in the same protein or other proteins. This survey will provide essential links between the structure and function of the protein. In my opinion, this is a valuable survey work and is suitable for publication in the International Journal of Molecular Sciences after the authors have addressed the following comments and questions:

Specific comments:

1)     Line 214 - This method is flexible and can be used for all soluble proteins -  is this a limitation for membrane-bound proteins or membrane protein solubilized into detergents?

2)     Line 240 - The CD is more time-consuming than DSC, taking a few hours to collect the data. – I disagree with this statement. Many new advanced CD instruments can finish temperature ramping experiments in less than an hour. However, I agree that the instruments are expensive and require more resources, like liquid nitrogen or helium gas.

3)     Line 301 – correct it  - “blotted as a slope”

4)     What is the significant delta Tm value for ligand-bound and unbound protein?

Comments on the Quality of English Language

minor typo error

Author Response

This manuscript, entitled “Fluorescence-based protein stability monitoring – a review,” by Gooran and Kopra, reviewed the fluorescence-based low instrument and expertise demand techniques to study protein stability related to structure and function. Also, they compared differential scanning fluorimetry (DSF) to isothermal chemical denaturation (ICD). A descriptive survey like this helps to find protein stability in a different environment to compare with its counterpart by creating mutation in the same protein or other proteins. This survey will provide essential links between the structure and function of the protein. In my opinion, this is a valuable survey work and is suitable for publication in the International Journal of Molecular Sciences after the authors have addressed the following comments and questions:

We sincerely thank the Reviewer for positive evaluation of our manuscript and for offering helpful comments to help us improve the manuscript. We have carefully considered each comment made by the Reviewer and fully addressed each comment made by the Reviewer as described point-by-point below:

1)     Line 214 - This method is flexible and can be used for all soluble proteins -  is this a limitation for membrane-bound proteins or membrane protein solubilized into detergents?

This is very good question and we now see that the sentence gives a wrong view about the method applicability. DSC indeed can be used for membrane proteins and what was the original idea of the sentence is that DSF for example is not really fully universal even for all soluble proteins. We have explained this now in more details and clarified that indeed DSC is membrane protein compatible.

2)     Line 240 - The CD is more time-consuming than DSC, taking a few hours to collect the data. – I disagree with this statement. Many new advanced CD instruments can finish temperature ramping experiments in less than an hour. However, I agree that the instruments are expensive and require more resources, like liquid nitrogen or helium gas.

I agree that our sentence is too simplified. Both CD and DSC instruments have become faster during the last years and also the most advanced DSC measurement can be performed in about an hour and with automation even faster. We have now added more discussion on this to state improvement related to instrumentation.

3)     Line 301 – correct it  - “blotted as a slope”

The correction has been made.

4)     What is the significant delta Tm value for ligand-bound and unbound protein?

That is important and interesting question, which is very hard to answer. ΔTm is highly dependent on the protein studied and the nature of the bound ligand, as ΔTm is not always something you can estimate only from information like affinity. Significant change also is dependent on the method used, as if the variation is large, higher shift is needed to make it significant. Typically, already 1-2 °C shift can be observed reliably and thus it can be considered significant enough, and in case of some interactions this is highest which can be obtained. However, in some cases, like with covalent binders, ΔTm might be tens of degrees, and thus giving only one value for the significant shift is difficult. We have added some discussion relate to this in the manuscript.